# A Comparison of the Epidemiology, Clinical Features, and Treatment of Acute Osteomyelitis in Hospitalized Children in Latvia and Norway

**DOI:** 10.3390/medicina57010036

**Published:** 2021-01-04

**Authors:** Elise Evja Thingsaker, Urzula Nora Urbane, Jana Pavare

**Affiliations:** 1Faculty of Medicine, Riga Stradins University, LV-1007 Riga, Latvia; elisethingsaker@gmail.com (E.E.T.); Jana.Pavare@rsu.lv (J.P.); 2Sørlandet Sykehus Helseforetak, 4604 Kristiansand, Norway; 3Department of Paediatrics, Riga Stradins University, LV-1004 Riga, Latvia

**Keywords:** acute osteomyelitis, children, antibacterial therapy, *Staphylococcus aureus*

## Abstract

*Background and objectives*: Paediatric acute osteomyelitis (AO) may result in major life-threatening and limb-threatening complications if not recognized and treated early. The management of AO may depend on local microbial prevalence and virulence factors. This study compares the approach to paediatric AO in hospitals in two countries—Latvia and Norway. *Materials and Methods:* The study includes patients with AO hospitalized in the paediatric department in the Norwegian hospital Sørlandet Sykehus Kristiansand (SSK), in the period between the 1st of January 2012 and the 31st of December 2019. The results from SSK are compared to the results of a published study of AO in patients hospitalized at the Children’s Clinical University Hospital (CCUH) in Riga, Latvia. *Results:* The most isolated pathogen from cultures in both hospitals was *S. aureus* (methicillin-sensitive). The lower extremity was the most affected body part (75% in CCUH, 95% in SSK), the main clinical symptom was pain (CCUH 92%, SSK 96.6%). Deep culture aspiration was most often taken intraoperatively in CCUH (76.6%) and percutaneously in SSK (44.8%). Oxacillin was the most applied antibiotic in CCUH (89.4%), and Cloxacillin in SSK (84.6%). Combined treatment with anti-Staphylococcal penicillins and Clindamycin was administered in 25.5% and 33.8% of CCUH and SSK patients, respectively. The median duration of the intravenous antibacterial treatment in CCUH and SSK was 15 and 10 days, respectively, and a switch to oral therapy was mainly made at discharge in both hospitals. The median total duration of antibiotic treatment was 25 days in CCUH and 35 days in SSK. 75% of CCUH and 10.3% of SSK patients were treated surgically. Complications were seen in 47% of patients in CCUH and in 38% in SSK. *Conclusions:* The transition to oral antibacterial treatment in both hospitals was delayed, which suggests a lack of criteria for discontinuation of intravenous therapy and could potentially contribute to longer hospitalization, higher cost of treatment and risk of complications. The use of more invasive techniques for deep culturing and significantly more common surgical interventions could possibly be linked to a higher complication rate in AO patients treated at the Latvian hospital.

## 1. Introduction

Paediatric musculoskeletal infections are a wide spectrum of disorders that may result in major life-threatening and limb-threatening complications if they are not recognized and treated early [1,2,3]. Acute haematogenous bone and joint infections in children may clinically manifest as osteomyelitis, septic arthritis, a combination of osteomyelitis and septic arthritis, or pyomyositis [3]. Acute haematogenous osteomyelitis (AHO), an inflammation and destruction of a bone caused by bacterial seeding through bacteraemia, is the most common among paediatric musculoskeletal infections [4,5]. The annual incidence of childhood AHO can vary from 1.94 to 8 in 100,000 in developed countries [6,7,8] to a much higher rate in developing countries and populations [9,10,11]. Around 50% of AHO cases occur within the first 5 years of life [12], and it is twice as common in boys than in girls [6,13]. 

In the case of AHO, the bacteria mostly seed in metaphyses of the bones due to abundant but slow blood supply [14]. The femur (23–29%), tibia (19–26%), and humerus (5–13%) are the most commonly affected bones [10,15], constituting approximately two thirds of all cases of AHO [16]. The causative microorganisms can vary in different populations, where they depend on vaccination programmes and regional antibacterial susceptibility, and techniques used for microbial culturing and identification can also affect the prevalence of isolated bacteria [17]. The aetiology of AHO also varies with age. The most commonly isolated microorganism from children of any age has been, for decades, *Staphylococcus aureus* [17,18,19]. *Kingella kingae* is a major cause of paediatric AHO in children younger than 4 years, in whom this pathogen can be isolated more frequently than *S. aureus* if molecular diagnostic methods are applied [18,20]. *Neisseria gonorrhea* is a possible causative agent in neonates and sexually active adolescents. In neonates, group B *Streptococcus* and enteric bacteria can also be isolated [16]. The role of *Haemophilus influenzae*, type B, formerly the second most common causative microorganism of AHO in children, and *Streptococcus pneumoniae*, is decreasing due to intensive vaccination [16,19,21]. Other rare isolated microorganisms may include Group A *Streptococcus*, *Neisseria meningitidis*, *Fusobacterium necrophorum*, and *Salmonella spp*. [17,22]. 

Early diagnosis and treatment are crucial for limiting the risk of complications [2,3,23], though recognition may be challenging due to the varying and sometimes non-specific presentation in paediatrics [1,3]. The initiated antimicrobial treatment is almost always empiric and depends mainly on local epidemiologic data an antimicrobial susceptibility [3,17]. Obtaining a microbial culture, such as a blood culture (though yielding results in only up to 40% of cases), and, in the case of periosteal pus, a bone sample via percutaneous aspirate, before initiation of antibiotic treatment is important but should not delay treatment [3,15], as it may help to identify the virulence factors of the causative microorganism and indicate the preferable choice of antibiotics and duration of treatment. The incidence of Methicillin-resistant *S. aureus* (MRSA) causing community-acquired AHO is increasing in the United States [24,25,26], while a necrotizing toxin, Panton–Valentine leucocidin (PVL), producing *S. aureus*, has been increasingly identified in bone and joint infections in Europe [27,28]. According to clinical practice guidelines created by the European Society for Paediatric Infectious Diseases (ESPID), antimicrobial therapy should cover methicillin-sensitive *S. aureus* (MSSA), for which 1st generation cephalosporins or anti-staphylococcal penicillins are usually chosen, and include antimicrobials (such as clindamycin) that are effective against MRSA in regions with a higher prevalence of the bacteria. In children younger than 4 years, adding third-generation cephalosporines may be considered to cover *K kingae*, and antimicrobials covering gram-negative flora should be added in newborns [3,29]. Empiric antibiotics should be switched to a targeted treatment once the causative agent is identified.

The recommended duration of antibiotic treatment in most cases of AHO is 3 to 4 weeks [3], though the applied strategies depend on the severity and the causative microorganisms (such as MRSA or PVL-producing *S. aureus*), and may vary in different countries [16]. Historically, AHO was treated with a prolonged intravenous strategy, while in recent years an early switch to oral antibiotics (in 2 to 7 days) in otherwise healthy children has been proven as equally effective and sparing of complications related to prolonged intravenous (IV) access [3,30,31], though factors such as the presence of fever, inflammation markers, virulence factors of microorganisms, and the improvement of symptoms are considered as well [3]. There are studies that show that the adherence to a short-duration IV treatment guidance is low, mostly due to concerns about the caregiver’s compliance to oral therapy at home [32]. 

Several studies show that in 90% of cases AHO in children, especially if treated within the first few days of symptoms, a conservative therapy with antibiotics is effective, and surgical intervention is not required [30,33]. There is, however, no consensus on the necessity, extent, and timing of surgical procedures [3,15]. In developed countries, AHO rarely results in fatal consequences [13,34], though a delay in antimicrobial treatment and highly virulent microorganisms can lead to devastating complications, such as abscess formation, deep vein thrombosis, septic embolism, sepsis with multiorgan failure, chronicity, impaired longitudinal bone growth, and deformity of the extremity [10,16]. 

Studies show that acute haematogenous osteomyelitis is an evolving disorder in terms of epidemiology, virulence factors of causative microorganisms, and management strategies, including methods of culturing, choice, and duration of antimicrobial therapy, as well as the application of a surgical approach. Therefore, local descriptive studies of AHO in the paediatric population can provide more insight into these aspects of the disease. The aim of this study is to clarify and compare the main causative agents of AHO and their antibacterial susceptibility, to evaluate the clinical signs, diagnostic, and treatment strategies and complications of acute osteomyelitis (AO) in paediatric patients in two countries—Latvia and Norway.

## 2. Materials and Methods

This is a retrospective observational study that includes patients hospitalized in the paediatric department in the Norwegian hospital Sørlandet sykehus Kristiansand (SSK) in the period between the 1st of January 2012 and the 31st of December 2019. The data were extracted from the medical records available in the archive of the hospital in question, with permission from the head of the department of paediatrics and the data protection officer. The data were compared with the published results of a similar study conducted in the Children’s Clinical University Hospital (CCUH) in Riga, Latvia: “*Epidemiology and Antibacterial Treatment of Acute Hematogenous Osteomyelitis in Patients Hospitalized at Children’s Clinical University Hospital in Riga, Latvia*” [35], including paediatric patients with acute osteomyelitis admitted to CCUH between the 1st of January 2014 and the 31st of December 2017.

In SSK, cases with the following codes of diagnosis from the 10th revision of International Statistical Classification of Diseases and Related Health Problems (ICD-10) were selected for the study:M86.0 Acute Haematogenous OsteomyelitisM86.1 Other acute osteomyelitisM86.2 Subacute OsteomyelitisM86.8 Other Specified Osteomyelitis, Brodie abscessM86.9 Unspecified Osteomyelitis

In CCUH, cases with the following ICD-10 codes were included [35]:
M86.0 Acute haematogenous OsteomyelitisM86.1 Other Acute OsteomyelitisM86.5 Other Haematogenous Osteomyelitis*

* The Latvian version of the ICD-10 of this code does not apply to chronic osteomyelitis alone. 

In both settings, patients with the following criteria were excluded [35]:Patients older than 18 years or younger than 7 days;Patients with comorbidities that may impair the immune response and are related to an increased infection risk: acquired or congenital immunodeficiency, use of immunosuppressants, patients undergoing chemotherapy, chronic illnesses; cystic fibrosis, diabetes mellitus, malignancies, myopathies, chronic kidney or liver insufficiency, or multiple genetic abnormalities;Secondary osteomyelitis after open fractures or surgery.The following data were collected and analysed in both hospitals [35]:Demographic data of patients;Causative microorganisms and their antibacterial susceptibility;Clinical presentation, including general and location-specific symptoms;Performed investigations and their results;Management strategy—duration of hospitalization, antibacterial treatment applied in the hospital and route of administration; antibacterial treatment and duration recommended after discharge; surgical interventions;Treatment outcomes and complications.

The obtained data were analysed using Microsoft Excel (version 16.44) and IBM SPSS (version 26) data analysis software, and descriptive statistical methods were applied. 

The study was a retrospective observational study which had no impact on any clinical interventions or outcomes. The patients’ identities were not exposed in the research paper or in the database. The study was approved by the data protection officer and Director of research in SSK (date of acceptance: 17 January 2020; code: 19/09958-4-552), and the Regional Committees for Medical and Health Research Ethics (date of acceptance: 6 November 2019; code: 64904), which accepted the study without necessity to take further action.

## 3. Results

### 3.1. Description of the Study Site (Sørlandet Sykehus Kristiansand)

SSK is a medium-sized regional hospital with a population of 155,000 affiliated inhabitants. In the Paediatrics Department there are 15 beds, and approximately 85% of hospitalizations are due to acute pathologies. The average duration of hospitalization for paediatric patients is approximately 48 h.

### 3.2. Study Population, Clinical Data, and Location of Inflammation

In total, 32 patient medical records qualified for analysis according to the selected ICD-10 diagnoses within the selected time frame. Three patients were excluded due to the exclusion criteria (one patient was immunosuppressed due to glucocorticoid therapy, one patient had secondary osteomyelitis after surgery, and one patient had died, and therefore the medical records were unavailable). The medical records of the remaining 29 patients with the diagnosis of acute osteomyelitis were analysed. All data recorded and used for analysis can be found in Appendix A. Of all included patients, 65.5% (*n* = 19) were boys and 34.5% (*n* = 10) girls. This is comparable to the result in the Children’s Clinical University Hospital of Riga (CCUH) with 64% (*n* = 60) boys out of a total of 102 patients [35].

Table 1 illustrates the demographic data of the study population of SSK, with the number of patients diagnosed with acute osteomyelitis each year, including gender distribution. The male to female ratio was 1.9:1, which is close to the CCUH ratio, of 1.7:1 [35].

The CCUH study reported that the number of acute osteomyelitis cases increased with age [35]. The study population of SSK showed similar tendencies, with only 3.4% (*n* = 1) of cases of osteomyelitis among children under 3 months of age, 41.3% (*n* = 12) within the age group of 3 months to 5 years, and the majority of patients (55.5% (*n* = 16)) diagnosed in the age group within 5 years to 18 years of age. Table 2 compares the age distribution of patients with osteomyelitis at CCUH and SSK. 

The referrals of the patients differed in the two study groups; this is most likely due to the different healthcare systems in the two countries. In CCUH, most patients were self-referred (56%), and only a small proportion of patients were referred by a general practitioner or an outpatient surgeon (13%). 31% of patients were referred from another hospital [35]. In SSK, as much as 75.9% (*n* = 22) were referred by a general practitioner or another specialist, 17.2% (*n* = 5) were self-referred, and 6.9% (*n* = 2) were transferred from another hospital. 

The average duration of illness before hospital admission at SSK was 11.3 days, ranging from 1 day to 112 days. The mean duration of symptoms prior to admission to CCUH was 3 to 4 days [35]. The patients who were later diagnosed with chronic multifocal recurrent osteomyelitis (CMRO) were the ones with the longest duration of symptoms prior to admission, due to atypical and unclear clinical pictures. Excluding these patients, the mean duration of symptoms prior to admission was 5.3 days in SSK. 

Pain was the most frequently reported symptom in both study populations, where it was characteristic of 93% (*n* = 88) of patients in CCUH and 96.6% (*n* = 28) at SSK. The second most common symptom was a limitation in the range of motion. In CCUH, swelling and hyperaemia were reported in 67% (*n* = 63) and 38% (*n* = 36) of cases, respectively, while in SSK they were noted in 34.5% (*n* = 10) and 20.7% (*n* = 6) of patients. 55.2% of patients had fever in (*n* = 16) at SSK, and 72.5% (*n* = 67) in CCUH. Tachycardia was present in nine Latvian and two Norwegian patients, while 45% (*n* = 42) of patients in CCUH and 37.9% (*n* = 11) in SSK had tachypnoea. The clinical symptoms of patients in SSK are listed in Table 3.

In patients of both hospitals, acute osteomyelitis occurred mostly in the lower extremity, the most frequently affected bones being the tibia, femur, and pelvis. In some patients, the joints, mostly in the lower extremity (hips, knees, feet), were also affected. The site of infection and the affected joints in both study populations are compared in Table 4 and Table 5. 

### 3.3. Laboratory and Imaging Diagnostics

All patients in SSK and CCUH underwent routine laboratory investigations upon admission. The most commonly performed tests were full blood counts, leucocytes, CRP, ESR, IL-6, kidney function tests, and electrolytes. For 51% of the study population of CCUH and for 82.8% of selected patients in SSK the leukocyte count was within the normal range at the time of admission. In CCUH, IL-6 was elevated in 72% (*n* = 68) of the studied patients, and this inflammatory marker was not used in SSK. The CRP was elevated in 86% of CCUH cases (38 of 46 tests) and in 81.5% of cases in SSK (22 of 27 tests). In CCUH, ESR was analysed upon admission in four cases and was elevated in all of them [35]. In SSK, ESR was assessed in 18 cases and was elevated in 17 of them. 

The diagnosis of acute haematogenous osteomyelitis was in most cases clinically suspected at the time of admission. Likewise, antibiotic therapy was initiated early, before radiological confirmation, in both hospitals. 

The most sensitive and frequently used radiological method for establishing the diagnosis in both hospitals was MRI. The confirmation by imaging started from day 3 of hospitalization. In all patients in SSK, MRI was performed at some point in the disease course, and in 89.6% of cases this was the confirmatory method. In two cases scintigraphy, and in one case aspiration biopsy was the confirmatory method. 62.1% (*n* = 18) of patients underwent an ultrasound examination, 17.2% (*n* = 5) patients had a CT scan, and scintigraphy was performed in 6.9% (*n* = 2) of patients.

The use of X-ray imaging was markedly different in the two hospitals. In CCUH all patients underwent an X-ray examination of the affected region at least once during hospitalization [35], while in SSK only 51.8% (*n* = 15) of the patients had an X-ray done, either ambulatory (*n* = 4) or on the first day of illness (*n* = 11).

### 3.4. Microbiology Investigations

In SKK, 89.6% (*n* = 26) of patients with acute osteomyelitis had blood cultures taken, compared to 63.8% (*n* = 60) in CCUH. The results in 73.2% (*n* = 19) of samples in SSK and in 55% (*n* = 33) of CCUH samples were negative. Table 6 demonstrates the results of the blood cultures of osteomyelitis patients in the Latvian and Norwegian hospitals. One sample was positive with multiple bacteria in CCUH, compared to zero samples in SSK.

No data were available on the intraoperative cultures (pus, tampons, synovial fluid) taken in the Norwegian hospital (given that only three patients underwent surgery), as opposed to the Latvian hospital, where intraoperative cultures were performed in 76.6% (*n* = 72) of cases, 15% of which were negative. The results from the intraoperative cultures in CCUH are listed in Table 7 [35]. In SSK, aspiration biopsies were performed in 44.8% (*n* = 13) of patients, 53.9% (*n* = 7) of which were negative. 66.6% (*n* = 4) of positive aspiration biopsies were positive for methicillin-sensitive *Staphylococcus aureus*, one was positive for *Klebsiella kingae*, and one for *Staphylococcus epidermidis* (possible contamination from skin flora). There were zero cases with detection of MRSA in the Norwegian hospital, and only one in the Latvian hospital. Table 8 exhibits the results from the aspiration biopsies in SSK. The results of identified microorganisms in cultures of patients in SSK according to age are shown in Table 9.

### 3.5. Management and Therapy

All patients in the CCUH study group received antibacterial treatment [35], whereas in SSK the percentage was 89.6% (*n* = 26). The three SSK patients who did not receive antibiotics were excluded from further analysis of antibacterial therapy. These patients presented an atypical clinical picture at the time of admission, and were later diagnosed with chronic multifocal recurrent osteomyelitis (CMRO). In 70.0% of the analysed cases in CCUH, antibiotic treatment was started on the first day of hospital admission. In SSK the time of initiating antibiotics ranged between the 1st and the 39th day of disease, and 49.4% of cases were given antibiotics on the first day. Antibiotic treatment was initiated on the second day of hospitalization for 11.7% of patients in CCUH and 22.8% in SSK. The administration of antibiotics was delayed to day 5 or later in five cases in CCUH and four cases in SSK. 

Monotherapy with one single agent was applied in 57% (*n* = 54) of CCUH patients, and in 73.1% (*n* = 19) of SSK patients. 43% (*n* = 40) of the analysed CCUH patients and 30.8% (*n* = 8) of SSK patients had combined antibiotic treatment of two or more antibiotics during the course of the illness. The most common combination therapy was Oxacillin and Clindamycin in CCUH, which was received by 25.5% of patients. In SSK, the most frequently given combination of antibiotics was Cloxacillin and Clindamycin, which accounted for 33.3% of all the combinations initiated (*n* = 12), not taking into consideration that some patients had multiple combinations tried during the illness. All in all, 50% (*n* = 4) of the patients receiving combination therapy (*n* = 8) were given the combination of Cloxacillin and Clindamycin at some point during the therapy. The most common initial treatment in the SSK patients was Cloxacillin monotherapy, which was given in 53.8% (*n* = 14) of cases. 

The most frequently applied antibiotic in CCUH was Oxacillin, which was received by 89.4% of patients [35]. Cloxacillin was the most used antibiotic in children with AO in SSK. It was received by 84.6% of patients. The antimicrobials used for treatment were switched in 77% of CCUH cases, and in 80.8% of cases in SSK. The most common reason for this was the unavailability of the oral form of the initiated intravenous antibiotic. Figure 1 reflects the antibiotics used for the treatment of acute osteomyelitis during hospitalization in SSK. 

Most patients in both studies received antibiotic treatment intravenously during hospitalization (92.3% in SSK). The intravenous treatment was changed to oral mostly at the day of discharge in both hospitals. The duration of antibiotic treatment in CCUH during hospitalization varied between 5 and 35 days, with a median of 15 days [35]. In SSK it varied between 2 and 65 days, with a median of 10 days. For four of the SSK patients in the study, this information was not documented.

In CCUH, a 14-day course of oral antibiotics was prescribed after hospitalization in the majority of cases (52%) [35], whereas in SSK, the most frequently recommended duration of oral antibiotic therapy after discharge was 21 days. Cefuroxime was the most popular oral antibiotic prescribed for CCUH patients, which was received by 52.5% of patients. In SSK, the oral antibiotic of choice was Clindamycin, which was prescribed to 54.2% of patients. Two medical records in SSK and nine medical records in CCUH did not give clear information about the choice, duration, and chosen antibiotic for the ambulatory treatment. Figure 2 indicates the prescribed oral antibiotics after discharge from SSK.

The total duration of antibiotic treatment in the CCUH cohort varied between 5 and 50 days, and between 13 and 86 days in SSK. In CCUH the median was 25 days, and 35 days in SSK. 

75% (*n* = 70) of the CCUH patients were treated surgically, whereas in SSK a surgical intervention was performed in only 10.3% (*n* = 3) of patients. The mean duration of hospitalization was similar for the two hospitals, 13 days in CCUH and 14.3 days in SSK.

### 3.6. Complications

Complications in patients treated for acute osteomyelitis in SSK differed comparing to the patients treated in CCUH both in frequency and range of complications. In total, complications were seen in 38% (*n* = 11) of the SSK patients, and 47% (*n* = 44) in CCUH patients. The two most common complications in SSK were abscess formation (45.5% (*n* = 5)) and chronicity of inflammation (45.5% (*n* = 5). There was one patient with failed antibiotic treatment, and therefore prolonged hospital stay and disease. The complications in CCUH included sepsis, septic arthritis, pneumonia, synovitis, phlegmon, pathological fractures, paresis, deep vein thrombosis, bursitis, abscess formation, and necrotizing fasciitis [35]. 

## 4. Discussion

Acute osteomyelitis is a serious bacterial infection with changing epidemiology and evolving approach to diagnosis and management. Though great efforts have been made to reach a consensus regarding the development of evidence-based guidelines for the diagnosis and management of musculoskeletal infections in children [1,3], little evidence was obtained from randomized control studies with large patient populations [3], as acute osteomyelitis in children living in developed countries is rare [6,7,8]. Rather, the consensus largely relies on expert opinions, case series, and cohort studies [1,3,10]. Deviations from clinical practice guidelines, such as the recommended duration of intravenous antibacterial therapy, have been observed due to the clinical judgement of physicians as to what is more appropriate for their patients [32]. Furthermore, the treatment and outcomes may be depending on local microbial prevalence and virulence factors, and thus the local peculiarities of these variables should be studied thoroughly.

This paper discusses the comparison of epidemiology, clinical presentation, diagnosis, management, and outcome in children with acute osteomyelitis treated in hospitals of two high-income European countries. Though the clinical presentation, causative microorganisms, and principles of antibiotic treatment of acute osteomyelitis were generally similar in both study populations, a few differences regarding diagnostic imaging, microbial culturing, and surgical intervention were identified. 

While in both CCUH and SSK the marked prevalence of male over female patients with acute osteomyelitis was similar to that described in most studies [6,10,13], only the study population of SSK reflected the classical tendency of a large proportion of patients being younger than 5 years, whereas in CCUH the majority of patients were older. In SSK, acute osteomyelitis was rarely found in adolescents. 

In CCUH, patients were admitted earlier in the course of the disease (3 to 4 days after developing symptoms, which is similar to reports elsewhere [5]) than in SSK (5.3 days on average). Patients of CCUH were most often self-referred, as opposed to SSK, where three quarters of the patients were referred by an general practitioner or a specialist after an ambulatory consultation, as a hospital specialist advice is usually unavailable in Norway prior to a visit to primary care. In CCUH, the proportion of patients in whom antimicrobial therapy was initiated on the day of admission was also higher than in SSK. As the outcome of acute osteomyelitis is time-sensitive [1,2,3], a greater emphasis on the early recognition of osteomyelitis should be placed, and the necessary access to healthcare should be rapidly provided for patients with suspected musculoskeletal infections. 

The most common symptoms at presentation—pain, loss of range of movement, tachypnoea, ill appearance, and fever—matched those reported in most studies [3,10,12,17]. Fever was more common in patients presented to CCUH than those admitted to SSK. As described in most sources [7,15,16,17,21,36], the infection mostly occurred in the lower extremity, with the tibia being the most commonly affected bone in SSK, and the femur in CCUH. 

The application of routine laboratory investigations was similar in both study sites. The normal leukocyte counts in most patients of both hospitals confirmed that haematological parameters should be interpreted with caution [37,38,39]. C-reactive protein level, which has been characterized as a useful marker for diagnosing AHO [5,39,40], was elevated in the vast majority of patients both in CCUH and SSK. ESR, another good marker for diagnosing AHO, though rising more slowly (reaching its peek on day 5–7 compared to 48 h for CRP) [5,40], was elevated in almost all patients. 

The diagnosis of acute osteomyelitis was clinically suspected upon admission, and antibiotics were initiated before the radiological confirmation of the diagnosis in both study groups. The use of conventional X-ray imaging was markedly different between the two hospitals—while all CCUH patients had an X-ray done at some point of the disease, this method was used in only a little more than a half of the patients in SSK (either ambulatory or in the 1st day of illness). Plain radiographs are considered as a necessary baseline test for patients with bone and joint infections [3]. Though they are only used in 20% of cases, X-ray images are useful for ruling out differential diagnoses such as trauma or malignancy [41]. Although it is known that no changes in conventional radiographs are expected at less than 7 to 10 days of the disease, and the sensitivity and specificity of early radiographs are low for diagnosis of osteomyelitis when compared to MRI [16,42,43], follow-up X-ray imaging can be useful in monitoring the response to antibacterial treatment, and an inexpensive way to rule out the chronicity of osteomyelitis [39,44]. MRI has been considered as the most informative diagnostic method for acute osteomyelitis, showing changes starting from day 3 of illness [3,41,45,46] and providing information on soft tissue involvement, including abscess formation, and the identification of a co-existing joint pathology [39]. MRI was performed in all SSK patients and was confirmatory of the diagnosis in 89.6% of the patients. In CCUH, the methods of conformation were mostly MRI and ultrasound scan [35]. An US scan is more readily available than MRI and can identify signs of osteomyelitis days earlier than plain radiographs [43]. However, the bone marrow cannot be viewed via a US scan, and the latter may therefore fail to identify acute osteomyelitis at early stages [39]. Ultrasound imaging has a lower sensitivity and specificity than MRI or bone scintigraphy [42], though it is invaluable in cases with metallic foreign bodies or other cases where MRI is contraindicated [42,43]. In addition, a US scan can be used for guidance when performing deep tissue cultures, pus drainage, or biopsies [47]. 

Blood cultures, though recommended for all patients with suspected acute osteomyelitis [3], were taken in less than two thirds of the patients in CCUH. In SSK, the method of choice for deep tissue culturing was the use of imaging-guided aspirations, and no intraoperative samples were cultured. In contrast, in CCUH, the former were never performed, and intraoperative material was cultured instead in three quarters of the cases. Of these deep cultures, more than a half were negative in SSK, whereas only 15% in CCUH. The role of intraoperative sample culturing is controversial in children with mild osteomyelitis without subperiosteal abscess, as it may lead to surgery-related adverse events and may have no therapeutic impact on the outcome [3,48]. Interventional radiology-guided culturing may be used instead as a less invasive option for obtaining a deep culture. Though studies report deep culturing as the only means of obtaining a microbiological means in just around one third of cases, it has proven helpful in choosing a more targeted and narrower-spectrum antibacterial treatment [48].

In agreement with most studies [10,17,18,19], Methicillin-sensitive *Staphylococcus aureus* was the most frequently isolated microorganism from both blood and deep cultures in both hospitals. No samples yielded MRSA in SSK, while in CCUH one sample did. The samples were not analysed for Panton–Valentine leukocidin (PVL) in *S. aureus*, but a study of CCUH patients with infections caused by *S. aureus* conducted between 2006 and 2008 has shown a 75% prevalence of PVL-positive isolates [49]. *Kingella kingae*, despite being identified as one of the main causes of acute osteomyelitis in very young children [18,20], was found in only one blood culture and bone aspiration sample in SSK in a child younger than 5 years. 

Though most patient cases included in the study predate the publication of the ESPID guidelines on bone and joint infection, which recommend that empirical antibacterial treatment should include, in most cases, antimicrobials covering MSSA (first- or second-generation cephalosporins or anti-Staphylococcal penicillins) [3], the choice of antibacterial therapy was similar. Oxacillin (second generation penicillin, beta lactamase resistant) and Cloxacillin (second generation penicillin, beta lactamase resistant) were received by the vast majority of CCUH and SSK patients, respectively. Oxacillin and Clindamycin was the most given combination in the Latvian hospital, and Cloxacillin and Clindamycin was the most frequently given combination in the Norwegian hospital, though the guidelines recommend the inclusion of antimicrobials covering MRSA (such as Clindamycin, Rifampin, or anti-Staphylococcal beta lactams) only in areas with a high prevalence (more than 10 to 15%) of the microorganism. The current studies on the prevalence of MRSA show a lower prevalence in both countries, though the reported incidence in Norway has increased significantly over the last decade [50,51,52,53], and only one MRSA sample was identified in CCUH. 

Most patients in both studies received intravenous antibiotic therapy initially, which was changed to oral treatment only at the day of discharge. The median duration of intravenous antibacterial treatment exceeded 10 days in both hospitals and was longer in CCUH than in SSK by five days, while the mean duration of the prescribed oral antibacterial treatment was longer for SSK than for CCUH by one week. The necessary duration of intravenous versus oral antibacterial treatment remains controversial. Nowadays, a shorter duration (2 to 7 days) of intravenous antibiotics prior to oral treatment is advocated for [3,30,31,54], as it may reduce the duration of hospitalization, and thus provide economic benefits, as well as prevent complications associated with prolonged intravenous access, while studies show that it can be equally effective [30,31]. However, facts such as the clinical improvement and decrease in inflammatory markers after the initial intravenous therapy, the presence or absence of complications, and the prevalence of methicillin-resistance and other virulence factors should be taken into account when deciding when the cessation of the intravenous treatment is possible [3]. In addition, in some cases, intravenous treatment may be prolonged as a result of parental compliance issues perceived by clinicians [32]. The factors influencing the switch to oral antibacterial therapy were not studied in either of the populations, and it is therefore not clear how many of the patients could have made the switch earlier. However, the fact that in almost all cases oral antibiotics were prescribed at discharge suggests that it may be the accepted practice of the day by the hospitals, and other factors may have been disregarded. In order to optimize the strategies for switch from IV to oral antibiotic treatment in both hospitals, a consensus for locally applicable guidelines should be made, stating objectively measurable factors that support either the early or late cessation of intravenous treatment. The factors allowing for an early switch to oral antibiotics include the absence of complications (such as sepsis, haematogenous spread of the infection to other organs (pneumonia) and deep vein thrombosis), the absence of fever for at least 48 h, a significant clinical improvement (reduced or absent pain and erythema), a significant reduction in inflammatory markers (30–50% decrease in CRP levels), the absence of virulent microorganisms (such as Salmonella, MRSA, or PVL+ *S. aureus*), and the absence of pathogens in the repeated blood culture (if initially present) [3,54]. 

The choice of antimicrobials for oral therapy is also debatable. The ESPID guidelines suggest that the oral antibiotics should match the previous treatment received intravenously [3]. Despite this, a change in antimicrobials was undertaken in three quarters of the cases in both hospitals, possibly due to the unavailability of the oral form of the initiated intravenous antibiotic drug. The suggested antimicrobials [3] for areas with low prevalence of MRSA are first- or second-generation cephalosporins (which was the practice in most of CCUH cases), or clindamycin as an alternative (practised in SSK). 

The study revealed significant differences regarding the surgical approach practised by both hospitals. Three quarters of the patients in CCUH underwent surgical therapy, compared to only one in ten of the SSK patients. There is evidence that antibacterial treatment can be effective in 90% of patients with acute osteomyelitis if diagnosed and treated early [3,15]. Surgical drainage may be required in case of subperiosteal pus collection, or abscess, but there is still no agreement on the indications, timing, and extent of the surgical intervention [3], nor on when it can be avoided (for example, when there are only early radiological signs of bone destruction and pus collection, and antibiotic therapy may be sufficient [37]).

This retrospective comparison study is not without its limitations. First of all, as acute osteomyelitis is rare in the paediatric population of developed countries [6,7,8], the number of patients included in the study is small. Secondly, the number of patients affiliated to SSK is smaller than that of CCUH, and thus the number of patients with acute osteomyelitis is also very small in comparison to the Latvian hospital, which limits the variability in patients to some extent. Thirdly a retrospective study is in itself a weakness due to the differences in documenting medical records from hospital to hospital and doctor to doctor. Nevertheless, meaningful information was obtained in the study, such as the causative microorganisms of acute osteomyelitis, the practices in microbial culturing and antibacterial therapy, the surgical interventions, and the outcomes of the disease. The results obtained in this study are comparable to those of other studies in the field. The study revealed some differences in the approach to acute osteomyelitis in children between a Latvian and a Norwegian hospital, including some deviations from the suggested clinical practice guidelines [3] in both hospitals (though it must be considered that a significant proportion of cases predate the publication of the guidelines). Hence, more studies in epidemiology and practices regarding musculoskeletal infections in children in other developed countries should be explored to gain more information and evidence in which to base internationally applicable guidelines.

## 5. Conclusions

The aetiology and clinical features of acute osteomyelitis were similar in both study populations and matched the findings of other studies. A tendency for a delayed transition to oral antibiotics was observed in both hospitals, which suggests a lack of established and applied criteria for the discontinuation of intravenous therapy, thus potentially increasing the duration of hospitalization, cost of treatment, and risk of complications. The surgical approach, though nowadays recommended only for complicated cases, was widely applied in the Latvian hospital, either for culturing or potentially therapeutic purposes, as opposed to the percutaneous aspiration biopsy used in the Norwegian hospital. The use of more invasive techniques could be linked to a higher complication rate in the acute osteomyelitis patents treated at the Latvian hospital.

## Figures and Tables

**Figure 1 medicina-57-00036-f001:**
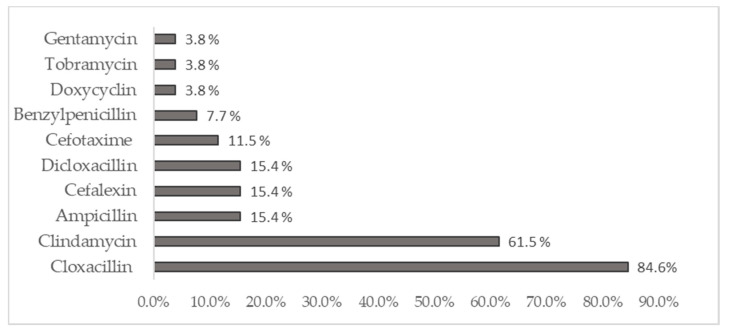
Antibiotics used for treatment of acute osteomyelitis during hospitalization in SSK.

**Figure 2 medicina-57-00036-f002:**
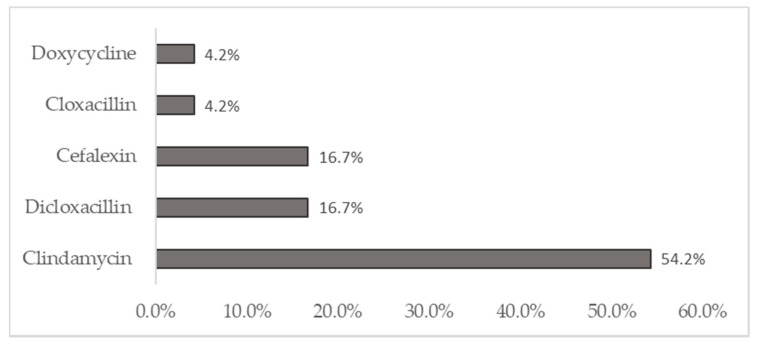
Antibiotics prescribed upon discharge from SSK.

**Table 1 medicina-57-00036-t001:** Number of patients diagnosed with acute osteomyelitis per year and gender distribution at Sørlandet Sykehus Kristiansand (SSK).

Year	Total No. of Patients	Male	Female
2012	1	1	0
2013	6	4	2
2014	8	4	4
2015	1	1	0
2016	2	1	1
2017	4	2	2
2018	4	4	0
2019	3	2	1

**Table 2 medicina-57-00036-t002:** The age distribution of patients with acute osteomyelitis at the Children’s Clinical University Hospital (CCUH) and SSK at the time of diagnosis.

Age Group	CCUH	SSK
<3 months	8.5% (*n* = 8)	3.4% (*n* = 1)
3 months to 5 years	25.5% (*n* = 24)	41.3% (*n* = 12)
5 to18 years	66% (*n* = 62)	55.5% (*n* = 16)

**Table 3 medicina-57-00036-t003:** Clinical symptoms reported in acute osteomyelitis patients in SSK.

Symptoms	*n* (%)
**Systemic symptoms**	
Fatigue, drowsiness	18 (62.1%)
Appears ill to health care professional	18 (62.1%)
Tachypnoea	11 (37.9%)
Refusal of food	11 (37.9%)
Tachycardia	2 (6.9%)
Irritability	9 (31.0%)
Poor weight gain	2 (6.9%)
Grunting, moaning	1 (3.4%)
**Localized symptoms**	
Pain	28 (96.6%)
Limited range of motion	13 (44.8%)
Swelling	10 (34.5%)
Hyperaemia, redness	6 (20.7%)

**Table 4 medicina-57-00036-t004:** Site of acute osteomyelitis in patients of CCUH and SSK: affected bones.

Affected Bone	CCUH *n* (%) [35]	SSK *n* (%)
Femur	27 (28.7%)	2 (6.9%)
Tibia	19 (20.2%)	11 (37.9%)
Metatarsal bones, wrist bones, ulna, metacarpal bones, spine	13 (13.8%)	2 (6.9%)
Pelvis	12 (12.7%)	5 (17.2%)
Humerus	9 (9.6%)	1 (3.4%)
Calcaneus	7 (7.4%)	5 (17.2%)
Mandibula, sternum, ribs, skull, maxilla, scapula, patella, talus	6 (6.4%)	4 (13.8%)
Fibula	5 (5.3%)	4 (13.8%)
Clavicle	3 (3.2%)	0 (0%)
Radius	1 (1.1%)	0 (0%)

**Table 5 medicina-57-00036-t005:** Affected joints in acute osteomyelitis patients in CCUH and SSK.

Affected Joint	CCUH *n* (%) [35]	SSK *n* (%)
Hip	11 (11.7%)	0 (0%)
Knee	3 (3.2%)	2 (6.9%)
Foot	3 (3.2%)	2 (6.9%)
Shoulder	2 (2.1%)	0 (0%)
Elbow	0 (0%)	0 (0%)

**Table 6 medicina-57-00036-t006:** Isolated bacteria in blood cultures of acute osteomyelitis patients in CCUH and SSK.

Bacteria	CCUH *n* (%) [35]	SSK *n* (%)
**Positive tests**	28 (45%)	7 (26.8%)
*Staphylococcus aureus*, methicillin sensitive	24 (40%)	5 (71.4%)
*Staphylococcus hominis*	2 (3.3%)	0
*Streptococcus pyogenes*	1 (1.6%)	1 (14.3%)
*Hemophilus parainfluenzae,* beta-lactamase negative	1(1.6%)	0
*Kingella kingae*	0	1 (14.3%)
**Negative tests**	33 (55%)	19 (73.2%)

**Table 7 medicina-57-00036-t007:** Results of intraoperative cultures of acute osteomyelitis patients in CCUH [35].

Bacteria Isolated	*n* (%)
*Staphylococcus aureus* (MSSA)	57 (79%)
*Streptococcus pyogenes*	2 (2.8%)
*Streptococcus viridans*	1 (1.4%)
Gram-negative cocci	1 (1.4%)
*Staphylococcus aureus* (MRSA)	1 (1.4%)
Negative	11 (15%)

MSSA—methicillin-sensitive *S. aureus*; MRSA—Methicillin-resistant *S aureus.*

**Table 8 medicina-57-00036-t008:** Results of aspiration biopsies from osteomyelitis patients in SSK.

Bacteria Isolated	*n* (%)
*Staphylococcus aureus*	4 (66.6%)
*Staphylococcus epidermidis*	1 (16.7%)
*Kingella kingae*	1 (16.7%)
Negative	7 (53.9%)

**Table 9 medicina-57-00036-t009:** Number of acute osteomyelitis cases and isolated microorganisms according to age in SSK.

Age	Number of Patients (% of Boys)	Isolated Microorganisms
<3 months	1 (100.0%)	Unknown
3 months to 5 years	12 (75.0%)	*Staphylococcus aureus* *Streptococcus pyogenes* *Kingella kingae*
5 to 11 years	12 (58.3%)	*Staphylococcus aureus* *Staphylococcus epidermidis*
12 to 18 years	4 (50.0%)	*Staphylococcus aureus*

## Data Availability

Data collected and analysed in the study are available in Supplementary file S1.

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
