# Peer review of "A Comparison of the Epidemiology, Clinical Features, and Treatment of Acute Osteomyelitis in Hospitalized Children in Latvia and Norway"

_medicina, 2021, doi:10.3390/medicina57010036_

Round 1
Reviewer 1 Report
In this work Thingsaker et al. compare the diagnosis and treatment approach of pediatric acute osteomyelitis in patients from a Norwegian hospital to similar data previously published on patients from a Latvian hospital. This study confirms several aspects previously reported for the Latvian cohort, highlighting several aspects of acute osteomyelitis handling that could be improved. However, I believe few major aspects could be improved:
- Could the authors clarify the rationale followed for the age stratification applied in their study? For the sake of comparison with the Latvian study it seems appropriate that analysis of the data in the same age groups is performed. However, I suggest the data collected is further analyzed taking into account other factors for age stratification. For instance, in the introduction, the authors state the aetiology of the disease varies with age, and give specific examples of the association between the predominant microorganisms isolated with specific age groups. I was wondering if further stratification taking into account average age for instance puberty or adolescence would impact the finding of increasing diagnosis of acute osteomyelitis with increasing age.
- Can the authors suggest further objective actions to be undertaken by clinicians in order to objectively evaluate and decide for withdrawal of i.v. antibiotics and initiation of oral treatment?
- Could the authors further comment on the differential methods of diagnostics used in both Hospitals, their advantages and disadvantages, and the impact of using one over other in disease diagnosis and evaluation of disease progression?
Author Response
Authors’ reply to Reviewer 1
Dear Reviewer,
We are delighted in your interest in our work, and we are very grateful for your suggestions. We tried to address all your commentaries to our best ability, and our response can be found below:
Reviewer commentary 1: Could the authors clarify the rationale followed for the age stratification applied in their study? For the sake of comparison with the Latvian study it seems appropriate that analysis of the data in the same age groups is performed. However, I suggest the data collected is further analyzed taking into account other factors for age stratification. For instance, in the introduction, the authors state the aetiology of the disease varies with age, and give specific examples of the association between the predominant microorganisms isolated with specific age groups. I was wondering if further stratification taking into account average age for instance puberty or adolescence would impact the finding of increasing diagnosis of acute osteomyelitis with increasing age.
Response to Reviewer commentary 1: Thank you for your commentary.
Per the suggestion in the commentary, we have added an additional table showing the number of cases and isolated microorganisms according to different age groups. The table can now be found in line 251 on page 7 of the reviewed manuscript. We stratified four different age groups – children younger than 3 months, age 3 months to 5 years, age 5 to 11 years, and 12 to 18 years:
Table 9. Number of acute osteomyelitis cases and isolated microorganisms according to age in SSK
|
Age |
Number of patients (% of boys) |
Isolated microorganisms |
|
<3 months |
1 (100.0%) |
Unknown |
|
3 months to 5 years |
12 (75.0%) |
Staphylococcus aureus Streptococcus pyogenes Kingella kingae |
|
5 to 11 years |
12 (58.3%) |
Staphylococcus aureus Staphylococcus epidermidis |
|
12 to 18 years |
4 (50.0%) |
Staphylococcus aureus |
It is evident that the Staphylococus aureus was identified in patients of all groups, and the patient in whom Kingella kingae was identified is younger than 5 years. The incidence of acute osteomyelitis in adolescents (children older than 12 years) in SSK was very low. Two comments addressing these findings can now be found in the Discussion section of the manuscript:
Lines 331-332 on page 10: “In SSK, acute osteomyelitis was rarely found in adolescents.”
Lines 378-380 on page 10: “Kingella kingae, despite being identified as one of the main causes of acute osteomyelitis in very young children [19, 21], was found in only one blood culture and bone aspiration sample in SSK in a child younger than 5 years.”
Reviewer commentary 2: Can the authors suggest further objective actions to be undertaken by clinicians in order to objectively evaluate and decide for withdrawal of i.v. antibiotics and initiation of oral treatment?
Response to Reviewer commentary 2: Thank you for your question.
We have taken your request for specific recommendations for the clinicians into account, thus the paragraph has been extended with the following text, which can be found in lines 425-433 on page 11 of the reviewed manuscript:
“In order to optimize the strategies for switch from IV to oral antibiotic treatment in both hospitals, a consensus for locally applicable guidelines should be made, stating objectively measurable factors that support either early or late cessation of intravenous treatment. The factors allowing an early switch to oral antibiotics include absence of complications (such as sepsis, hematogenous spread of the infection to other organs (pneumonia), and deep vein thrombosis), absence of fever for at least 48 hours, significant clinical improvement (reduced our absent pain and erythema), significant reduction in inflammatory markers (30 – 50% decrease in CRP levels), absence of virulent microorganisms (such as Salmonella, MRSA or PVL+ S. aureus), and absence of pathogens in the repeated blood culture (if initially present)”.
Reviewer commentary 3: Could the authors further comment on the differential methods of diagnostics used in both Hospitals, their advantages and disadvantages, and the impact of using one over other in disease diagnosis and evaluation of disease progression?
Response to Reviewer commentary 3: Thank you for your suggestion.
As requested, we have edited the discussion section of the manuscript, and the following sections describe the differences in diagnostic approach in both hospitals, their advantages and disadvantages:
Lines 360 – 374 on page 10:
“Although it is known that no changes in conventional radiographs are expected at less than 7 to 10 days of the disease, and the sensitivity and specificity of early radiographs are low for diagnosis of osteomyelitis when compared to MRI, follow-up X-ray imaging can be useful in monitoring response to antibacterial treatment, and an inexpensive way to rule out chronicity of osteomyelitis. MRI has been considered as the most informative diagnostic method for acute osteomyelitis, showing changes starting from day 3 of illness and also providing information on soft tissue involvement, including abscess formation, and identification of co-existing joint pathology. It was performed in all SSK patients and was confirmatory of diagnosis in 89.6% of the patients. In CCUH, the methods of conformation were mostly MRI and ultrasound scan. An US scan is more readily available than MRI and can identify signs of osteomyelitis days earlier than plain radiographs, however, viewing bone marrow via US scan is not possible, thus US scan may fail to identify acute osteomyelitis at early stages. Ultrasound imaging has lower sensitivity and specificity than MRY or bone scintigraphy, though it is invaluable in cases with metallic foreign bodies or other cases when MRI is contraindicated. In addition, US scan can be used for guidance when performing deep tissue cultures, pus drainage, or biopsy.”
Lines 380 – 383 on pages 10 and 11:
“The role of intraoperative sample culturing is controversial in children with mild osteomyelitis without subperiosteal abscess, as it may lead to surgery-related adverse events and may have no therapeutic impact on the outcome [4, 48]. Interventional radiology guided culturing may be used instead as a less invasive option for obtaining a deep culture.”
We opted not to comment of the use of CT scans and bone scintigraphy, as their role in diagnosis of acute osteomyelitis is not described in the publication on CCUH by Petruhina et al, thus any comparison would be inappropriate.

Reviewer 2 Report
The authors performed a retrospective observational study on pediatric patients with acute osteomyelitis comparing data from two institutions.
The article is well written and describes a useful "real-life" experience on the topic. I found especially interesting the description of the clinical management and the comparison between the two hospitals regarding the complications.
I only suggest correcting the use of abbreviations of "intravenous" in the text i.e. line 83 (i/v), line 87 (IV), etc...
Author Response
Authors’ reply to Reviewer 2
Reply to the Reviewer’s commentary:
Thank you for your assessment of our study, we greatly appreciate your approval.
We have made the requested corrections, and abbreviation “IV” is used for “intravenous” instead of “i/v”. The correction can be viewed in line 86 on page 2, while the appropriate abbreviation is used further in the text.

Round 2
Reviewer 1 Report
I appreciate the effort of the authors in addressing all the comments to their work, which I believe they addressed successfully. As a minor point, I cannot find at the moment a citation in the main text referencing the newly added Table 9, and this should be included.